# Sensory function and somatosensorial system changes according to visual acuity and throwing techniques in goalball players: A cross-sectional study

**Ayşenur Gökşen** [1]⊕ *, **Gonca İnce** [2]⊕

**1** Department of Physiotherapy and Rehabilitation, Faculty of Health Sciences, Tarsus University, Mersin, Turkiye, **2** Department of Coaching Education / Sport-Health Sciences, Faculty of Sports Sciences, Çukurova University, Adana, Turkiye

⊕ These authors contributed equally to this work.
* aysenurgoksen@tarsus.edu.tr

**Data Availability Statement:** All relevant data are within the manuscript and its Supporting Information files.

## Abstract

The somatosensory system is a complect sensory system that differentiates individual athletes. The aim of this study is to investigate the effect of visual acuity level on throwing technique, proprioceptive sense of the shoulder joint, light touch and two-point discrimination sense of the upper extremity, and sensory function (postural control and reaction time) in visually impaired goalball players. Goalball players who have different visual acuities B1 (unable to perceive light or recognize its shape); B2 (has a visual field of less than 5 degrees and can recognize shapes); B3 (visual field greater than 5 degrees and less than 20 degrees) participated in the study. The sensorial system was evaluated with proprioceptive sense of the shoulder joint and sensory tests (light touch and two-point discrimination sense of the dominant hand.). Sensory function (postural control and reaction time) was evaluated with the flamingo balance test, functional reach test, and pro-agility test. The goalball players' throwing technique was questioned. Seventeen male players, those aged 20–30 (20.8 ±3.9 years) who have been professionally engaged in goalball for at least three years (58.7–37.8 months) participated. Shoulder internal rotation joint position sense and the flamingo balance test were found to be different in the group with B1 visual acuity than in the group with B3 visual acuity (p = 0.042* and 0.028 respectively). There was no difference between groups with B1-B2 visual acuity (p = 0.394 and p = 0.065) and between groups with B2-B3 visual acuity (p = 0.792 and p = 0.931). There was no difference in the groups in terms of sensory tests and reaction time (p> 0.05). In goalball, joint position sense is related to throwing techniques. Although there is a general acceptance that other sensory systems should work harder to compensate for the sense of vision, fear of falling, athlete's branch year, sports year, muscle strength, and general physical condition of the athlete may affect the measurements made, especially in the dynamic position.

**Funding:** This work was supported by The Scientific and Technological Research Council of Turkiye under Grant 122C262. The funders had no role in study design, data collection and analysis, decision to publish, or preparation of the manuscript. There was no additional external funding received for this study.

**Competing interests:** The authors have declared that no competing interests exist.

## Introduction

Goalball is a paralympic sport that is played by visually impaired person and requires agility, balance, and good use of body biomechanics [1]. Goalball is also the universal language of social participation for the visually impaired. There are a limited number of studies in the literature regarding the participation of visually impaired individuals in sports activities [2]. Since manual positioning for visually impaired people is auditory and tactile, they must have more advanced and effective sensory sensitivity than sighted people. This enables them to overcome challenges and complete the goalball play successfully. Many different sensory systems such as vision, hearing, joint position sense (proprioceptive sense) and touch sense are called Somatosensory systems. This system is a personal complex sensory system. Vision, hearing, joint position sense, and other sensory organs play an important role in performing sports-specific skills [3, 4]. When the reception of visual stimuli is limited, the data flow decreases [5–9]. The sense of touch, where the movement is perceived first, the senses created by the movement on the joint, the processing of information in the somatosensory cortex, and the reaction process are effective in the goalball player's ability to learn a motor skill and perform [10]. The factors affecting this process can be examined in two ways: sensory and motor. Sensory factors can be interpreted by the somatosensory system and motor factors by muscle strength. In this study, the factors affecting throwing technique will be examined from a sensory perspective [10–12]. Reaction time and postural control are accepted as a measure of sensory function [12–14]. Grouping individuals according to visual acuity will more clearly reveal the effect of vision on the somatosensory system.

The aim of this study is to investigate the effect of visual acuity level on throwing technique, proprioceptive sense of the shoulder joint, light touch and two-point discrimination sense of the dominant hand, and sensory function (postural control and reaction time) in goalball players.

## Materials and methods

This project was carried out as a descriptive study in the cross-sectional survey model. This cross-sectional study was conducted between 3 December 2022 and 3 May 2023 in Çukurova University Sports Sciences Faculty Laboratory. Necessary permissions for the study were obtained from the Non-Invasive Clinical Research Ethics Committee of Çukurova University Faculty of Medicine. (Date: 02.12.2022, decision no. 50.). Written informed consent was obtained from all individual participants included in the study. The need for informed consent was not waived by the ethics committee. The authors did not have access to information that could identify individual participants during or after data collection. The study was carried out in accordance with the Declaration of Helsinki.

### Participants

This study was carried out on male players with different visual acuities between the ages of 18 and 26 who have been professionally involved in goalball for at least three years in Adana Region visually impaired sports clubs. Goalball players are divided into 3 groups according to their visual acuity: B1 (unable to perceive light or recognize its shape). B2 (has a visual field of less than 5 degrees and can recognize shapes). B3 (visual field greater than 5 degrees and less than 20 degrees). The required sample size was computed by G*Power 3.1.9.4 Software. The analysis of variance (ANOVA) repeated measures was used within factors having the effect size considered at 0.25. The non-sphericity correction was calculated as 1 and correlation among repeated measures was determined as 0.5 considering three groups. The predicted sample of 15 participants were adequate for a statistical power 80% and a level of 5% alpha margin

of error. Considering the duration of the study, we estimated that there might be a loss in participants to fallow up, therefore 17 samples were included in the study [1, 15–17].

Inclusion criteria for participants in this study:

In this study, visually impaired male players aged 18–30 who have been playing on the goalball team for at least three years and who regularly practice goalball were included.

Exclusion criteria for participants in this study:

- Patients with a deviation of more than 30 degrees according to the Fukuda balance test were not included in the study because it may affect the postural control results of the study.

- Players who experienced serious injury within the first 6 weeks after the start of the study were not included in the study.

- Players with health problems (e.g., cancer, arthritis, heart disease, lung disease, and neurological disease) were not included in the study.

- Players with a disability other than the visually impaired were not included in the study.

- Players who regularly engage in other sports branches, except for goalball, were not included in the study.

Demographic characteristics, somatosensory system (shoulder joint proprioception sensation and sense of touch), postural control (static and dynamic), and reaction time were evaluated in this study.

## Procedure of testing

All tests were completed at Çukurova University Faculty of Sports Sciences Laboratory. First, a form was filled out in which the goalball players 's throwing technique, dominant extremity injury history, sports age, branch year, and other demographic characteristics (age, height, weight) visual acuity were questioned. Secondly, shoulder joint position sense and sensory tests of the dominant hand was evaluated. Afterwards, the Flamingo balance test and functional reaching test were performed. A warm-up protocol was applied before the pro agility test. Finally, the pro-agility test (reaction time) of the goalball players was evaluated. For each test, three familiarization trials were made, and each test was repeated three times, and the average was taken. The test procedure is shown in Fig 1.

## Shoulder joint proprioception sense test

The joint position sense of the shoulder internal and external rotation was evaluated by measuring in the supine position with an inclinometer device. The reliability rate of the

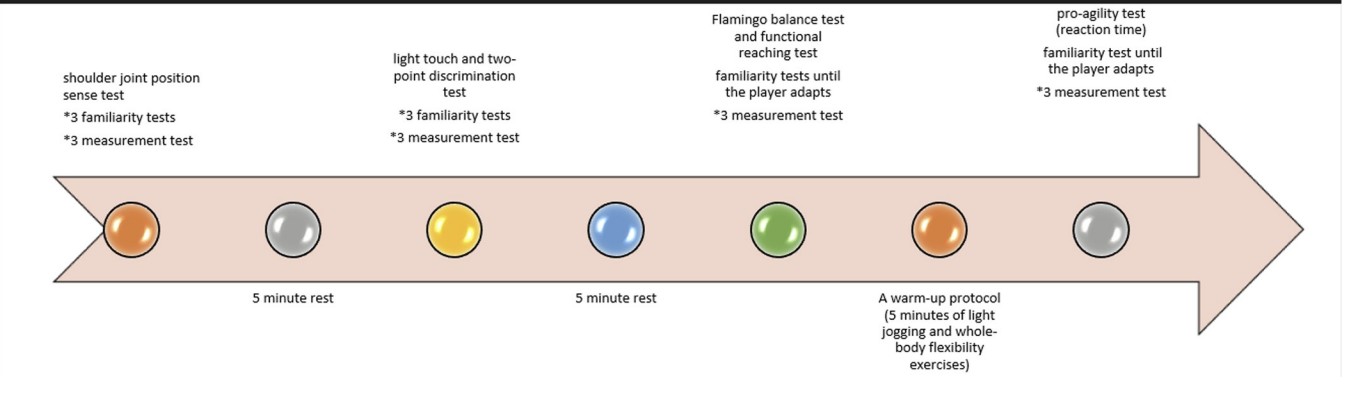

**Fig 1.**

inclinometer device for joint position sense is reported to be % 95–99 [18]. A passive shoulder joint repositioning test was used to test shoulder proprioception of the dominant extremity. Before starting the test, we securely strapped the inclinometer to the subject's wrist using Velcro straps. Then, the shoulder and elbow were kept in 90° abduction and flexion, respectively, while the range of motion was evaluated in terms of internal rotation (IR) and external rotation (ER) (Fig 2). The subject was instructed to actively rotate his arm to the extreme ends of movement in both the IR and ER directions. We then calculated repositioning (target) angles based on the maximum angle achieved by each subject. For this study, two target angles were equivalent to 90% of IR and 90% of ER ROM. In the standard application of this test protocol, the perception of joint movements is required by turning off the sense of vision, so we did not need additional modifications for this test. The difference between the goalball players' target angle and the angle which player repositioned was recorded as the declination angle (target angle-repositioned angle). The declination angle was used for statistical analysis [19, 20]. (See Fig 2).

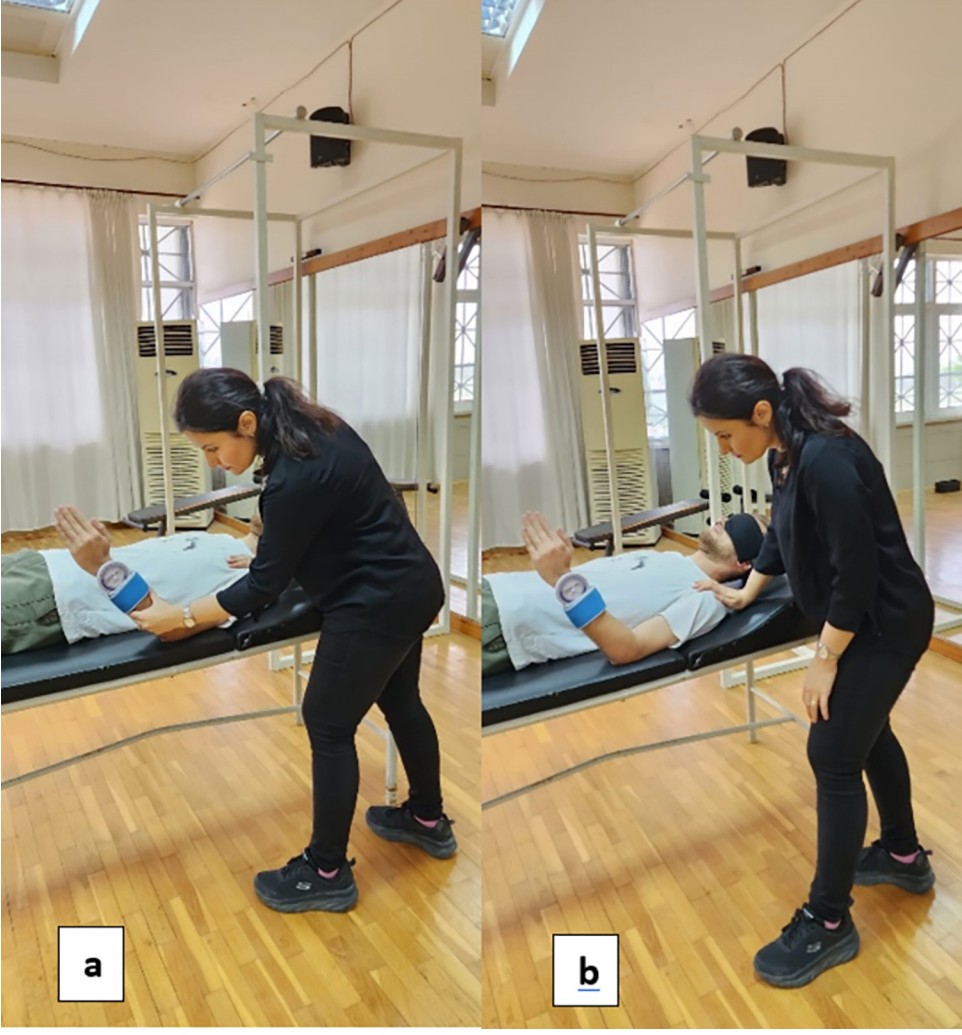

**Fig 2.**

## Sensory tests

The sense of touch was evaluated under two different headings: light touch and two-point discrimination, from the shoulder joint and palmar aspect of the hand.

Tactile sense was evaluated with Semmes-Weinstein Monofilament test and two-point discrimination tests. Before the tests, a familiarity trial was conducted. During the Semmes-Weinstein monofilament test, dermatome sites on the hand were considered. The filament was pressed and held for 1 second until it bent laterally to the dermatome area to be evaluated. The goalball players were asked if he felt the pressure of the monofilament. Each evaluation was made three times, and 15 seconds waited after each application. Verbal notifications from goalball players were recorded. All goalball players used eye patches during the test. Since this test is normally performed with the eyes closed, we did not need to make any additional modifications for the visually impaired. The only modification we implemented was using a vision patch to equalize visual acuity. Constant skin contact was applied to all goalball players s until the filaments became severely curled or removed. There are also studies in the literature that apply this test to visually impaired individuals [21–23]. (See in Fig 3).

1 mm precision calliper (Duratech™ TA-2081) was used to evaluate the two-point discrimination. The midpoint of the palmar aspect of the hand was also marked to standardize the test site. For the palmar face of the hand, the line drawn at the midpoint of the palmar face of the hand is the target. With the calliper, it was applied with equal pressure from both ends. Goalball players were instructed to report if they felt they had lost a point or two. Goalball players were instructed to report if they felt a point or two. Evaluations were made on the dominant extremity [24]. Visual acuity was equalized using a vision patch during testing [25].

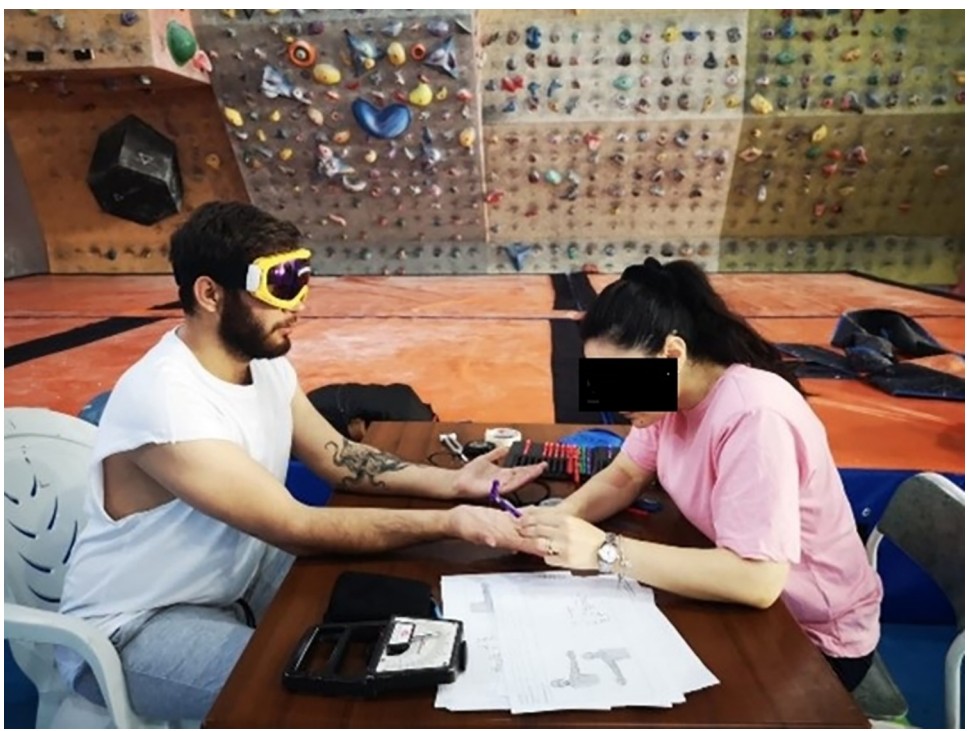

**Fig 3.**

## Sensory function

Sensory function was assessed by measuring reaction time and postural control.

**Reaction time.** Reaction time was evaluated with adapted pro-agility test. The basic reason we chose the pro-agility test was that it was more suitable for the agility movements used by goalball players in the game. It was also a testing tool that could be done more easily with an eye patch. Goalball players warmed up for 10 minutes before the test to reduce the risk of injury.

**Warm-up protocol (10 minutes).** 5-minute jogging (100–120 heart rate) and 5-minute stretching exercises

Stretching exercises 1. Forward and backward head stretch 2. Sideways head stretch 3. Chest and shoulders stretch 4. Deltoid muscle stretch 5. Triceps muscle stretch 6. Overhead stretch 7. Lateral trunk muscle stretch 8. Arched back stretch 9. Leg extensor and pelvic flexor stretch 10. Spinal twist stretches 11. Paravertebral muscle stretch 12. Loosen-up stretch 13. Upper back prayer 14. Double knee-to-chest stretch [26].

Pro-agility test practise:

In the pro-agility test, a 20-meter test area was created by placing the funnels 5 yards (4.57 m) to the left and right of the starting line. A photocell gate was placed at the start and finish line to obtain repetitive transit times.

Modifications of pro-agility test for goalball players:

A table tennis court screen [each:75cm(H)x140cm(L)] was placed around the test field to prevent the player from leaving the field. Moreover, a rough, stable surface was glued to the ground, on which the player would move. Thus, the player could stay in the testing area. (See in Fig 4).

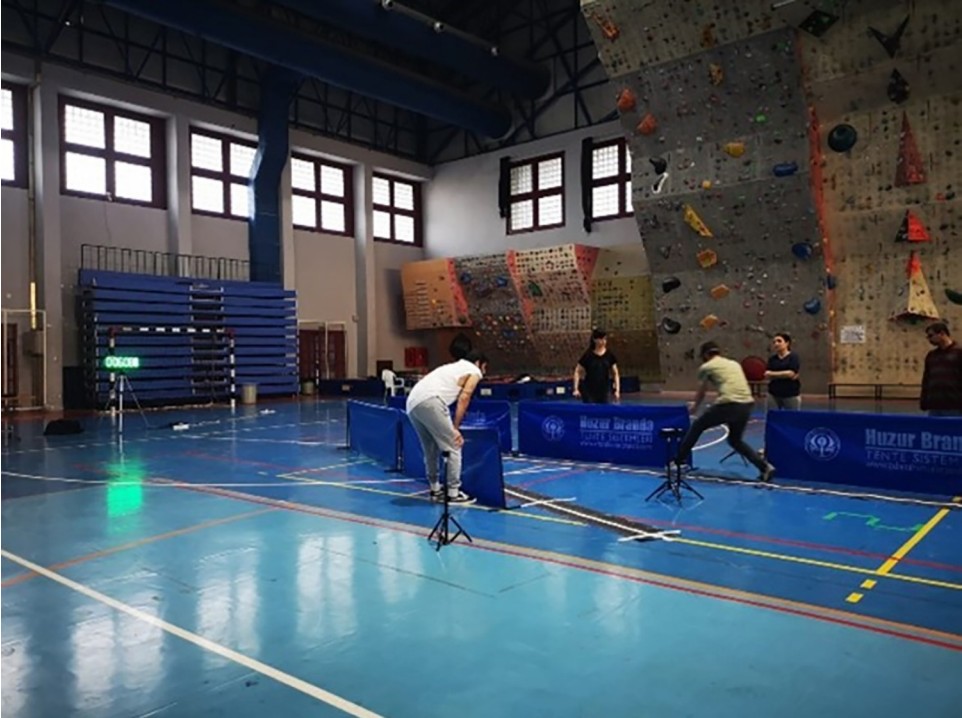

**Fig 4.**

The goalball player who wore eye bant took their place at the starting line before the test started. When he was ready, he passed through the photocell and started the stopwatch. He moved forward in the test field with touching the table tennis court screen. Then he touched the cone on the right and then the cone on the left, crossing the starting line and ending the test. The time seen in the photocell was recorded, see Fig 2.

Before the test, for the reaction time test, a familiarity assessment was carried out together with the person administering the test. The testing field and boundaries were introduced to the goalball players. The rules of the test were explained. The familiarity test was repeated until the goalball players had understood it. The reaction time test application was repeated three times, and the average of three measurements was used. If the goalball players left the testing area during the test, the test was terminated [24].

**Postural control.** Postural control was evaluated statically and dynamically with the eyes closed. Stopwatch, tape measure and wooden block were used for the test.

### Flamingo Balance Test

The Flamingo Balance Test, whose validity and reliability are determined by Tsigilis et al., is used to evaluate the static balance [27]. A wooden block consisting of 50 cm long, 4 cm high, and 3 cm wide beams and 2 short, 15 cm long, 2 cm wide beams was used for the Flamingo balance test. The familiarity test was repeated until the goalball players understood the test. The arms were spread out to the sides and used for balance. The player balanced on one leg barefoot, bending his free leg 90 degrees at the knee and keeping his foot close to the hip. When the goalball players were ready, the stopwatch was started by cutting the support, and time was stopped whenever the balance was disturbed. The time the goalball players were able to maintain his balance position was recorded. Modification of the test for visually impaired individuals: The goalball players was helped to get on the wooden block and take the position specified in the protocol. When the goalball players felt ready, their hand was released. All players wore eye patches during the test [28, 29].

### Functional reach test

Each player was asked to stand on the side of the hard wall to which a tape measure was affixed. The preferred arm was kept straight with the elbow, and the arm was raised until 90 degrees of flexion were achieved at the shoulder. The starting point of the longest finger was marked as the starting point of the individual. The individual reached the last point he could reach by leaning forward to follow the tape measure without disturbing the parallelism of the elbow, hand position, and arm with the ground. This end point was marked, and the difference from the starting point was recorded in centimetres (cm). The test was repeated three times, and the best value was recorded.

**Modification of the test.** The goalball players' transportation and positioning to the testing area was done by the investigator. He was asked to lie forward until he lost his balance [30, 31]. All players wore eye patches during the test.

### Statistical analysis

Data were analyzed within the IBM SPSS Statistics ver. 24.0 (IBM Co., Armonk, NY, USA). The statistics of descriptive variables were reported using mean and standard deviation (SD). The normality of distribution was tested by the Kolmogorov–Smirnov test. Nonparametric tests were selected according to the statistical evaluation. Chi-square tests were performed to compare categorical data with each other. To analyse the significance of difference of the outcomes between the three groups, Kruskal-Wallis test was conducted. The Kruskal-Wallis test

used to evaluate the change in numerical data such as joint position sense and reaction time according to visual acuity. When the Kruskal–Walli's test result was found to be statistically significant, pairwise comparisons were made with the Mann Whitney u test. Spearman correlation test was also used to evaluate the relationships between parameters. The internal consistency was evaluated using Cronbach's alpha coefficient, and a value of ≥0.70 was considered acceptable internal consistency. The non-probability purposive sampling method was used in the selection of the sample. Participants were selected according to the inclusion criteria. Inclusion criteria were reached through judgmental sampling methods.

**Measurement's reliability.**   The same researcher carried out all measurements. 3 repeated measures were used to assess intratest reliability. Intraclass correlation coefficient was calculated to determine whether there was a significant change between initial and repeated measurements.

## Results

A total of forty-five goalball players from three different sports clubs interested in goalball in the XXX region were reached. Ten players were female, and fifteen players did not participate in the training regularly, so they were excluded from the study. One goalball player was excluded due to an acute wrist injury. Two goalball players did not come to the evaluation. In conclusion, seventeen male players (B1 = 6, B2 = 6, B3 = 5) participated in the study. The mean age of goalball players is 20.8 (SD = 3.9) years. The mean body mass index of the participants was 22.6 (SD = 1.9) kg/m$^2$. It was noted that the players included in the study were interested in goalball for an average of 58.7±37.8 months. When the injury history of the players was taken, it was recorded that there was a total of 5 players with a history of mild orthopaedic injuries such as sprains and injuries. The demographic data of the players included in the study are presented in Table 1. There is no missing data in the study.

Shoulder internal rotation joint position sense and the flamingo balance test were found to be different in the group with B1 visual acuity than in the group with B3 visual acuity (p = 0.042* and 0.028* respectively). There was no difference between groups with B1-B2 visual acuity (p = 0.394 and p = 0.065) and between groups with B2-B3 visual acuity (p = 0.792 and p = 0.931), (Table 2). There was no difference in the groups in terms of sensory tests (p > 0.05), (Table 3).

It was found that the players included in the study preferred the straight throw more than the rotary throw. The agility, shoulder internal, and external joint position sense of the

**Table 1. Demographic data of goalball players.**

|  |  | B1 (n = 6) | B2 (n = 6) | B3 (n = 5) | Total (n = 17) | p |
|---|---|---|---|---|---|---|
| Age (year) |  | 22.6±4.67 | 19.6±2.30 | 20.0±4.1 | 20.8±3.9 | 0.329 |
| Cause of disability | Congenital | 6 | 6 | 5 | 17 | 0.378* |
|  | Postnatal | 0 | 0 | 0 | 0 |  |
| Body mass index |  | 23.0±2.73 | 22.5±1.76 | 22.3±1.76 | 22.6±1.9 | 0.662 |
| Sports age (month) |  | 54.0±50.62 | 86.0±59.47 | 86.0±59.47 | 68.4±46.1 | 0.429 |
| Branch year (month) |  | 54.0±50.62 | 58.5±43.2 | 64.8±10.7 | 58.7±37.8 | 0.429 |
| Sports Injury History |  | 2 | 1 | 2 | 5 | 0.676* |
| Throwing techniques | Frontal | 4 | 3 | 4 | 11 | 0.580* |
|  | Spin | 2 | 3 | 1 | 6 |  |

p = Kruskal Wallis test; p* = chi square test; n = number; B1. Goalball players have limited perception of light and are not able to recognize any distance or any direction
B2. Goalball players can recognize shapes up to a certain visual field. B3. The visual field and visual acuity of the goalball players are above a certain level.

**Table 2. Evaluation of sensory function according to visual acuity (reaction time and postural control).**

|  |  | B1 | B2 | B3 | p |
|---|---|---|---|---|---|
| **Reaction Time** |  | 12.4±2.9 | 10.0±2.8 | 12.3±3.3 | 0.260 |
| **Postural Control** | **Flamingo balance test** | 4.9±1.7 | 2.6±1.7 | 2.21±0.2 | 0.028* |
|  | **Functional reach test** | 37.8±12.5 | 18.1±5.1 | 28.2±15.5 | 0.166 |

P = Kruskall-wallis test B1. Goalball players have limited perception of light and are not able to recognize any distance or any direction B2. Goalball players can recognize shapes up to a certain visual field. B3. The visual field and visual acuity of the goalball players are above a certain level.

individuals who preferred the rotating shot were found to be better than the individuals who preferred the straight shot. It was found that sports age, branch year, and sense of reward did not affect Throwing techniques (p>0.05), (Table 4). In addition, it was found that the level of visual acuity did not affect throwing techniques (p = 0.691).

The assessment revealed good agreement between the two measurements (Table 5), demonstrating the reliability of the measurements. Table 5 shows the intraobserver reliability using intraclass correlation coefficient for measurements of shoulder joint position sense and sensory function.

## Discussion

This study evaluated goalball players with different visual acuities in terms of perception, joint position sense, dynamic and static balance, perception time, and throwing techniques and revealed their differences. The most important result of this study is that B1 (Goalball players have limited perception of light and are not able to recognize any distance or any direction) goalball players are more successful in static balance and shoulder joint position sense than B3 (The visual field and visual acuity of the goalball players are above a certain level) goalball players. It is similar in terms of sensory tests, dynamic balance, and reaction time tests. It was found that B1 goalball players adapted better to tests performed in a static position with eyes closed compared to B3 goalball players. The results in dynamic measurements may have been affected by the intense fall and crash anxiety in individuals with vision loss. This fear may be more intense in B1 goalball players than in B3 players due to the experiences gained. Fear of

**Table 3. Evaluation of senses (insert position sense, tender sense) according to visual acuity.**

|  |  |  | B1 | B2 | B3 | p |
|---|---|---|---|---|---|---|
| **Shoulder Joint Position Sense** | **Internal Rotation** |  | 2.0–3.9 | 4.5–5.6 | 8.4–3.8 | 0.042* |
|  | **External Rotation** |  | 2.8–3.8 | 4.5–5.6 | 4.6–6.9 | 0.263 |
| **Sensory tests** | **Two-Point Discrimination** | **1. Fingertip** | 4.6–1.8 | 4.3–1.5 | 5.0–4.2 | 0.792 |
|  |  | **2. Fingertip** | 4.8–1.6 | 4.1–1.3 | 5.0–4.2 | 0.662 |
|  |  | **3. Fingertip** | 4.6–1.8 | 4.8–2.1 | 5.0–4.2 | 0.792 |
|  |  | **4. Fingertip** | 4.6–1.8 | 4.6–1.3 | 5.0–4.2 | 0.792 |
|  |  | **5. Fingertip** | 5.1–1.6 | 5.0–4.2 | 5.0–4.2 | 0.662 |
|  |  | **Palm** | 6.6–1.9 | 6.5–3.6 | 6.5–3.6 | 0.662 |
|  | **Light Touch** | **base of thumb** | 2.8–0.3 | 2.9–0.5 | 2.8–0.5 | 0.804 |
|  |  | **the ulnar side of the hand** | 2.8–0.3 | 2.6–0.2 | 2.8–0.5 | 0.257 |
|  |  | **Palm** | 2.8–0.3 | 2.9–0.5 | 2.8–0.5 | 0.791 |

B1. Goalball players have limited perception of light and are not able to recognize any distance or any direction B2. Goalball players can recognize shapes up to a certain visual field. B3. The visual field and visual acuity of the goalball players are above a certain level.

**Table 4. Factors affecting throwing techniques.**

| Factors | | | Throwing techniques (n = 17) | | |
|---|---|---|---|---|---|
| | | | Frontal (n = 11) | Spin (n = 6) | p |
| Sports Age (month) | | | 57.8–36.6 | 88.00–58.6 | 0.202 |
| Branch Year | | | 57.8–36.6 | 60.50–43.6 | 0.387 |
| Agility test | | | 11.8±1.9 | 11.08–3.8 | 0.034* |
| Shoulder Internal Rotation Joint Position Sense | | | 7.5–6.8 | 1.81–1.7 | <0.001** |
| Shoulder External Rotation Joint Position Sense | | | 4.6–5.7 | 3.50–2.6 | 0.015* |
| One Leg Standing Test | | | 3.2–1.8 | 3.44–1.9 | 0.971 |
| Functional Reach test | | | 27.0–15.2 | 29.8–11.9 | 0.608 |
| Sensory Tests | Two-point Discrimination | 1. Fingertip | 4.4–2.7 | 5.00–2.3 | 0.642 |
| | | 2. Fingertip | 4.3–2.6 | 5.16–2.0 | 0.610 |
| | | 3. Fingertip | 4.4–2.7 | 5.50–2.5 | 0.880 |
| | | 4. Fingertip | 4.5–5.1 | 2.65–2.3 | 0.883 |
| | | 5. Fingertip | 5.3–2.0 | 4.63–2.6 | 0.757 |
| | | Palm | 8.0–4.4 | 5.72–3.0 | 0.206 |
| | Light Touch | base of thumb | 2.9–0.2 | 2.87–0.7 | 0.072 |
| | | ulnar side of hand | 2.9–0.2 | 2.59–0.4 | 0.184 |
| | | Palm | 2.8–0.3 | 2.9–0.3 | 0.073 |

Mann Whitney u test

p*<0.05

p**<0.001

falling causes movements to be more controlled and slower [32]. However, since we have not come across a scale in the literature that evaluates the fear of movement or fear of falling in visually impaired people, we only present our observations. A surprising result was that the sense of touch did not vary according to visual acuity. In individuals with vision loss, to compensate for the sense of sight, it is known that other sensory systems will be used more, and therefore the sense of touch is more developed than in sighted individuals [22]. Although the difference between sighted individuals and visually impaired individuals has been demonstrated, there is no study in the literature that evaluates changes in the sense of touch according to visual acuity. The visual acuity level may not have affected the somatosensory system enough to cause changes in sensory systems. On the other hand, the tactile test results of all goalball players may have been similar due to the positive effects of the sport on all sensory systems. Previous studies have shown that sport improves reflexes, senses, agility, balance, and proprioceptive senses in general [33–35]. As in other paralympic sports, all goalball players use eye patches in goalball. There is no obstacle classification in goalball [36]. However,

**Table 5. The interobserver reliability using intraclass correlation coefficient for measurements of shoulder joint position sense and sensory function.**

| | | | Cronbach alpha | ICC (95% CI) |
|---|---|---|---|---|
| Shoulder Joint Position Sense | Internal Rotation | | 0.98 | 0.94 |
| | External Rotation | | 0.99 | 0.98 |
| Sensory Function | Postural Control | Flamingo balance test | 0.79 | 0.56 |
| | | Functional reach test | 0.99 | 0.99 |
| | Reaction Time | | 0.93 | 0.82 |

Icc: intraclass correlation coefficient.

adaptations can develop in the somatosensory systems of visually impaired person individuals to adapt to changing conditions. In the study of Kimyon and Ince, it was emphasized that the goal-throwing performances of the goalball players s with better visual acuity were higher [2].

Therefore, it can be said that visual acuity in goalball is important for goalball players' performance. An eye patch is used to provide equal opportunity to the goalball players in the game of goalball. However, it is not clear how vision loss will create adaptations in other sensory functions related to performance. In sports with eyes closed, whether the group with low vision or the group with better visual acuity is more advantageous. Studies on this subject are also limited [2]. In a systematic review, the postural controls of visually impaired individuals were investigated. Although individuals with congenital blindness show structural changes in the cerebral cortex to compensate for the blindness, it has been reported that this does not lead to better postural control than individuals with typical development [37]. A few of the studies in the literature found that sports and exercise improved the postural control of blind individuals [4, 15, 34, 37].

In most studies in the literature, postural control assessments were performed with the eyes open. In our study, an eye patch was used for all evaluations. According to the postural control results of the study, it was seen that the shoulder joint position and static balance test results of the goalball players with less visual acuity were better. This situation supports the judgment that the visual acuity of the goalball players can affect their performance even if they use an eye patch.

Visually impaired athletes had a hard time with the Flamingo balance test and had difficulty adapting to the test. In the functional reaching test, they felt safer and adapted better because they were lying along a stable wall with a tape measure drawn. This situation was also reflected in the inter-measurement reliability tests. The reliability of the functional reaching test was higher than the flamingo balance test.

Likewise, the reliability of the reaction time test was lower than that of other tests. It can be said that in environments where the player does not have support material, this situation increases the athlete's movement anxiety and reduces the reliability of the test.

Goalball is a sport that requires quick manoeuvres from side to side and where agility and strength are important. Reaction time is a parameter that gives an idea about the functions of all the senses that directly affect the sport's skill. In the literature, the reaction time of visually impaired individuals is evaluated using the pro-agility test [38]. However, according to our observations, there is a need to adapt this test to visually impaired goalball players or to develop a new test specific to visually impaired goalball players s. According to a study, it is stated that visual acuity affects reaction time in sports because as visual acuity decreases, reaction time increases [39]. The important thing here is whether the evaluations are made using an eye patch. According to the evaluations made without using an eye patch, low vision constitutes a disadvantage. However, in our study, it was found that goalball players with low visual acuity were more advantageous in terms of static balance, according to the evaluation results using an eye patch. There is no difference between the dynamic balance test results according to visual acuity.

The shoulder internal rotation joint position was found to be more successful in the group with low visual acuity. It may have developed more in the group with less visual acuity since the internal rotation of the shoulder is used more during activities of daily living. The flow of information coming from the skin, muscles, joints, tendons, and tissues of the goalball players and the goalball players 's movement patterns according to this information flow Therefore, the quality of the shoulder joint's proprioceptive sense will affect both throwing performance and the risk of injury. Joint position sense in goalball players is a topic that has started to come to the fore in recent years, and there are limited studies in the literature [40]. Visual

communication provides the most important data in the proprioceptive process. It is assumed that other perceptions develop in more visually impaired individuals than in sighted person; however, there are a limited number of studies on this information in the literature. In a thesis study, it was determined that the knee joint proprioception of visually impaired goalball players was better than that of normal goalball players. Similar to the results of our study, the sense of shoulder internal rotation joint position was found to be better in the group with lower visual acuity than in the other group. In another study conducted on visually impaired goalball players in the literature, it has been reported that the knee joint position sense of visually impaired goalball players is better than that of sighted goalball players, and it has been suggested that this deficiency can be treated by training the sighted goalball players in the eyes closed position [14].

Although there are studies on balance and joint position in the literature, no study has been found about tactile sense. It is important to evaluate the sensitivity of mechanoreceptors in order to provide sufficient physiological evidence regarding the proprioceptive process. According to the results of our study, there was no difference between the presenting senses of the groups in terms of visual acuity. According to a study conducted on visually impaired children in the literature, it has been reported that visually impaired children use only tactile sense without the help of visual perception because they have lost their congenital vision; therefore, children with normal vision have better hand function test results [41]. In our study, however, no difference was found in terms of tactile, sensory, or visual acuity. Since the group evaluated in our study was made up of goalball players s, there may not have been a difference. Because it has been reported in previous studies that sports have positive effects on sensory function [38, 39].

Another variable examined in our study is throwing techniques. The main goal in goalball is to get the ball to the goal. It is possible sample for the goalball players to deliver the ball to the goal with a technical throw. Rotary throwing and straight throwing are two different throwing techniques. Although there is not enough evidence in the literature, the rotational shot provides acceleration and makes the shot go harder [15]. Rotational throwing is a difficult technique with a high goal rate, requiring more biomechanics [15]. The results of our study also supported this information. According to our results, goalball players s who prefer rotational throwing have better shoulder internal rotation joint position sense and agility.

## Conclusion

The visually impaired population has not been adequately studied in sports and exercise science. It is important to expand studies on goalball, which is a universal language of social participation for visually impaired individuals. This study was one of the rare studies that evaluated goalball players according to visual acuity. The results obtained from the study will contribute to the training to be given to increase the shots that result in goals and to improve the player's throwing technique in goalball, a paralympic sport. In the results of the study, shoulder internal rotation joint position sense was found to be related to shot preference in goalball. Goalball players, who have a better sense of joint position, prefer rotational throwing. Although it is thought that other sensory systems need to work harder to compensate for the sense of vision, fear of falling, fear of movement, athlete's branch year, sports year, athlete's muscle strength, and general physical fitness will affect the measurements made in the dynamic position.

### Limitation

The limited number of visually impaired individuals who regularly practice goalball also prevented the study from being conducted with a large sample size. The biggest reason for the limited number of participants is that all participants are active professional goalball players.

### Recommendations for future work

In this study, the somatosensory system was examined from a sensory perspective. In future studies, the somatosensory system can be investigated from a motor perspective. In other words, the effects of muscle strength and endurance on sensory function can be investigated.

This study was conducted on disabled athletes. In future studies, it can be investigated whether there are differences in the sensory systems between visually impaired individuals who are not athletes and healthy individuals.

Again, in future studies, the effects of proprioceptive and sensory training given to goalball players on the athlete's performance and injury risk can be investigated. Additionally, we would also be pleased to take part in a joint working group with researchers who want to support our ongoing work on these issues.

## Supporting information

**S1 Checklist. STROBE statement—Checklist of items that should be included in reports of observational studies.**
(DOCX)

**S1 Data. Data record of the study.**
(XLS)

## Acknowledgments

The authors acknowledge to goalball players who contributed to this study.

## Author Contributions

**Conceptualization:** Ayşenur Gökşen, Gonca İnce.

**Data curation:** Ayşenur Gökşen.

**Formal analysis:** Ayşenur Gökşen.

**Investigation:** Ayşenur Gökşen, Gonca İnce.

**Methodology:** Ayşenur Gökşen, Gonca İnce.

**Project administration:** Ayşenur Gökşen, Gonca İnce.

**Resources:** Ayşenur Gökşen, Gonca İnce.

**Supervision:** Ayşenur Gökşen, Gonca İnce.

**Validation:** Ayşenur Gökşen.

**Visualization:** Ayşenur Gökşen.

**Writing – original draft:** Ayşenur Gökşen.

**Writing – review & editing:** Ayşenur Gökşen.

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
