## [Decision Letter · Decision Letter 0]

30 Oct 2023

PONE-D-23-29101Specific changes in sensory function and somatosensorial system according to visual acuity and throwing techniques in goalball playersPLOS ONE

Dear Dr. Gökşen,

Thank you for submitting your manuscript to PLOS ONE. After careful consideration, we feel that it has merit but does not fully meet PLOS ONE’s publication criteria as it currently stands. Therefore, we invite you to submit a revised version of the manuscript that addresses the points raised during the review process.

Please submit your revised manuscript by Dec 14 2023 11:59PM. If you will need more time than this to complete your revisions, please reply to this message or contact the journal office at plosone@plos.org. Please include the following items when submitting your revised manuscript:A rebuttal letter that responds to each point raised by the academic editor and reviewer(s). You should upload this letter as a separate file labeled 'Response to Reviewers'.A marked-up copy of your manuscript that highlights changes made to the original version. You should upload this as a separate file labeled 'Revised Manuscript with Track Changes'.An unmarked version of your revised paper without tracked changes. You should upload this as a separate file labeled 'Manuscript'.

We look forward to receiving your revised manuscript.

Kind regards,

Monika Błaszczyszyn

Academic Editor

PLOS ONE

Journal Requirements:

This work was supported by The Scientific and Technological Research Council of Turkiye under Grant 122C262. The funders had no role in study design, data collection and analysis, decision to publish, or preparation of the manuscript.

5. We note that Figure 2 includes an image of a participant in the study.

Reviewers' comments:

Reviewer's Responses to Questions

**Comments to the Author**

1. Is the manuscript technically sound, and do the data support the conclusions?

Reviewer #1: Partly

Reviewer #2: Partly

2. Has the statistical analysis been performed appropriately and rigorously? 

Reviewer #1: Yes

Reviewer #2: No

3. Have the authors made all data underlying the findings in their manuscript fully available?

Reviewer #1: Yes

Reviewer #2: Yes

4. Is the manuscript presented in an intelligible fashion and written in standard English?

Reviewer #1: No

Reviewer #2: Yes

5. Review Comments to the Author

Reviewer #1: Dear Authors,

I added comments to specific parts of the manuscript PDF in yellow notes and yellow highlights in the text. Authors did not use STROBE checklist, They did not use PICOs method to describe aims what is highly recommended.

I hope they will be helpful for you to improve significantly the manuscript.

Please, underlign changes in the manuscript using different colour of fond, e.g., blue.

Kind regards,

Reviewer

Reviewer #2: Specific changes in sensory function and somatosensorial system according to visual acuity and throwing techniques in goalball players

General:

The study investigates an interesting phenomenon, that I believe would interest a good number of readers. The methods are interesting, and the visually impaired population is not well studied in sports and exercise science.

While the writing is okay, it can be improved in spots, and additional details are needed in a few areas.

Title:

Should the title read “…and THE somatosensorial system…”?

The title could also be slightly shortened to read “Sensory function and somatosensorial system changes according to visual acuity and throwing techniques in goalball players

Abstract:

The abstract is generally well-written (with the main exception of the very first sentence). However, I have some points for improvement and clarification.

The first sentence is quite clunky. I suggest re-writing to something like “The somatosensory system is a complect sensory system that differentiates individual athletes.”

Could the authors please briefly explain what ‘B1, B2, B3’ mean?

The number of significant digits I the participant age and experience should be reduced from “20.82” to 20.8, and “58.76-37.87” to 58.8-37.9. This should be done in the article text as well.

Im not sure why there is a star besides the p=0.042. I assume because it is statistically significant, but then why no star besides the 0.028?

Introduction:

Having an additional 1-3 sentences explaining goalball would be great.

There are a few examples of redundant sentences in the introduction. For example, the authors write about the somatosensory system affecting athletic success on lines 51-52, and then again on line 55. I personally would remove it from the first sentence of the introduction. Also please check throughout and try to simplify the introduction by avoiding redundancies or non-critical information.

The final sentence of the first paragraph should somehow be worked into the second paragraph instead.

Overall, the intro is good and interests me in the study.

Methods:

The levels of visual acuity should be explained in detail! What is B1, B2, B3 etc.? This is very important.

Line 127: is the 95-99 and ICC? I am not sure, please clarify/re-write.

Some form of reliability statistic is included for the shoulder joint proprioception sense. Could the authors provide similar information for the other tests?

It is great that the authors used effect size statistics, not just p-values.

The authors should explain why the Spearman correlation was used instead of the more common Pearson’s.

Results:

Apply the same feedback regarding significant digits that I mentioned in the abstract here.

While the authors mention Cohen’s D effect sizes in the methods, they are nowhere to be found in the results or discussion…

Discussion:

A ‘Discussion’ subsection title is missing.

The discussion is good, but would be further strengthened by the effect size statistics!

Please add a few ideas/directions for future research to the ‘limitations’ subsection.

Figures/tables:

An additional figure showing the ‘shoulder joint proprioception sense’ test would be welcomed.

Table 1 mentions that the ‘*’ is for the ‘chi square test’. But this test is not mentioned in the methods. What is it adding to the Kruskal Wallis test?

Table 4s legend says “p*<0.04. Should this be 0.05?

6. PLOS authors have the option to publish the peer review history of their article (what does this mean?). If published, this will include your full peer review and any attached files.

Reviewer #1: No

Reviewer #2: **Yes: **Dustin J Oranchuk

---

## [Author Response · Author response to Decision Letter 0]

1 Dec 2023

I would like to thank our esteemed referees and editors for their contributions.

• Reviewer #1: Dear Authors,

I added comments to specific parts of the manuscript PDF in yellow notes and yellow highlights in the text. The authors did not use the STROBE checklist, they did not use the PICOs method to describe aims what are highly recommended.

I hope they will be helpful for you to improve significantly the manuscript.

Please, underline changes in the manuscript using different colours of fond, e.g., blue.

Kind regards,

Entire sections of the article were reorganized based on comments noted in the text, with yellow notes and yellow highlights on specific sections of the draft PDF.

Line 26-29 the aim of the study at abstract 

Line 70-72 the aim of the study at introduction

Please, see the comments to this abstract above.

Could you rebuild this abstract to be more coherent?

Line 25-47

maybe adding a keyword: paralympics , would be useful?

Line 48

Individuals with visual impairments, according to APA style and firs-person language. I suggest to see: https://apastyle.apa.org/style-grammar-guidelines/bias-free-language/disability

The use of visually impaired person is recommended. Sample articles are attached.

1-Li, B., Yu, Y., & Hu, J. (2021). Applying the ICF-CY in visually impaired rehabilitation: a case report in China. Annals of palliative medicine, 10(3), 3459–3468. https://doi.org/10.21037/apm-20-312

2- Leissner, J., Coenen, M., Froehlich, S., Loyola, D., & Cieza, A. (2014). What explains health in persons with visual impairment?. Health and quality of life outcomes, 12, 65. https://doi.org/10.1186/1477-7525-12-65

3- Billiet, L., Van de Velde, D., Overbury, O., & Van Nispen, R. M. (2022). International Classification of Functioning, Disability and Health core set for vision loss: A discussion paper and invitation. British Journal of Visual Impairment, 40(2), 109-116. https://doi.org/10.1177/02646196211055954

According to the STROBE method of writing paper it is a good practice to add the type of the study method in the title of the paper. Please see STROBE checklist. It would be helpful to organize the paper paragraphs.

Title is revised. Added a file showing page numbers for STROBE.

Maybe would be better to write: Inclusion criteria for participants in this study?

Line 94-96

Evaluation - in general would be a good idea to describe the procedure of testing in this article.

Which test was the first, were there any breaks, how about everyday activities? Did participants have any warmup? etc.

Done. Line 110 Procedure of Testing

What does it mean “she” in this sentence?

Goalball player (all of gender is male). The expression "she" is wrong. Thank you to the referee. 

please add references of this test and clear description of the test and its modification

line 121-137

please add references of this test and clear description of the test and its modification

line 160-193

please add references of this test and clear description of the test and its modification

line 196-223

I don’t see ES in any table. Could Authors add to tables ES even for p > 0.05?

Line 238-241

Line 287-290

Line 337-344

Shoudln’t be written: ….was set at p value < 0.05.?

Line 278

my proposition would be to divide Results and Discussion.

Discussion should be full of results from diffeent studies, Authors opionions, analysis of resutls from their study and others resutls study. I would suggest to add more references, more Authors opinions and indicationa why results are important for GB players. There is lack of information of GB players and their results inconducted tests.

discussion was revised.

Line 290

please add below the table 3 the description of acronyms from the table 3

line 269

conclusions are not coherent with abstract’s conclusions. Could Authors rewrite conclusion in the abstract part?

Line 396

General:

The study investigates an interesting phenomenon, that I believe would interest a good number of readers. The methods are interesting, and the visually impaired population is not well studied in sports and exercise science.

While the writing is okay, it can be improved in spots, and additional details are needed in a few areas.

Title:

Should the title read “…and THE somatosensorial system…”?

The title could also be slightly shortened to read “Sensory function and somatosensorial system changes according to visual acuity and throwing techniques in goalball players

The title of the study has been changed 

Abstract:

The abstract is generally well-written (with the main exception of the very first sentence). However, I have some points for improvement and clarification.

The first sentence is quite clunky. I suggest re-writing to something like “The somatosensory system is a complect sensory system that differentiates individual athletes.”

The first sentence of my summary was changed as suggested by the referee

Could the authors please briefly explain what ‘B1, B2, B3’ mean?

An explanation of the abbreviations B1-B2-B3 was added to the summary, text, and below the tables.

Goalball players who have different visual acuities B1(unable to perceive light or recognize its shape); B2 (has a visual field of less than 5 degrees and can recognize shapes); B3 (visual field greater than 5 degrees and less than 20 degrees) participated in the study.

The number of significant digits I the participant age and experience should be reduced from “20.82” to 20.8, and “58.76-37.87” to 58.8-37.9. This should be done in the article text as well.

• The reduction requested by the referee regarding the mean and standard deviations was made. The p-value was not reduced because it was not recommended for p-values.

I'm not sure why there is a star besides the p=0.042. I assume because it is statistically significant, but then why no star besides the 0.028?

• Yes, I put a star to draw attention to the statistically significant one, and a star was added next to 0.028.

Introduction:

Having an additional 1-3 sentences explaining goalball would be great.

There are a few examples of redundant sentences in the introduction. For example, the authors write about the somatosensory system affecting athletic success on lines 51-52, and then again on line 55. I personally would remove it from the first sentence of the introduction. Also please check throughout and try to simplify the introduction by avoiding redundancies or non-critical information.

The final sentence of the first paragraph should somehow be worked into the second paragraph instead.

Overall, the intro is good and interests me in the study.

• Repetitive sentences with the same meaning in the introduction have been removed.

• The introduction started by adding a sentence about goalball: Goalball is a paralympic sport that is played by people with visual impairments and requires agility, balance, and good use of body biomechanics [1]. Goalball is also the universal language of social participation for the visually impaired.

Methods:

The levels of visual acuity should be explained in detail! What is B1, B2, B3 etc.? This is very important.

• Participants, Line 88-90

Goalball players are divided into 3 groups according to their visual acuity: 

B1 (unable to perceive light or recognize its shape). 

B2 (has a visual field of less than 5 degrees and can recognize shapes). 

B3 (visual field greater than 5 degrees and less than 20 degrees).

• Line 127: is the 95-99 and ICC? I am not sure, please clarify/re-write.

Some form of reliability statistic is included for the shoulder joint proprioception sense. Could the authors provide similar information for the other tests?

• These power analysis results. It was edited and written in the section related to the participants. Based on the sample study, the required sample size was calculated with the G*Power program. Line 90-93

• Since all measurements were made three times and averaged, it was not difficult to calculate inter-measurement reliability. It was stated in the findings with the appendix

• Table 5. The interobserver reliability using intraclass correlation coefficient for measurements of Shoulder Joint Position Sense and Sensory Function.

• Since sensory tests give sharper results and do not show variability, a single measurement record was reported, even though three measurements were made in accordance with the procedure. For example, if the patient cannot feel the red monofilament, he will not feel it even if we touch it 20 times. If the purple monofilament feels it, it will feel it with every touch.

It is great that the authors used effect size statistics, not just p-values.

The authors should explain why the Spearman correlation was used instead of the more common Pearson’s.

• Normal distribution tests were performed. The histogram was viewed. Since the data did not show a normal distribution, a Spearman correlation analysis test was used.

Results:

Apply the same feedback regarding significant digits that I mentioned in the abstract here.

While the authors mention Cohen’s D effect sizes in the methods, they are nowhere to be found in the results or discussion…

• The sample size required to carry out the study was added to the participants section of the method. Intermeasurer reliability tests were also added to the results. I apologize to the referees for the confusion.

• TABLE 5

Discussion:

A ‘Discussion’ subsection title is missing.

• The discussion section is separated from the conclusion. I think there was a misunderstanding regarding the spelling rules that I should give the discussion and the result together. I specifically thought I should make it combined.

 Please add a few ideas/directions for future research to the ‘limitations’ subsection.

*Recommendations for future work

Line 413-423 In this study, the somatosensory system was examined from a sensory perspective. 

 The discussion is good, but would be further strengthened by the effect size statistics!

• Line 337-344 Visually impaired athletes had a hard time with the Flamingo balance test and had difficulty adapting to the test. In the functional reaching test, they felt safer and adapted better because they were lying along a stable wall with a tape measure drawn. This situation was also reflected in the inter-measurement reliability tests. The reliability of the functional reaching test was higher than the flamingo balance test.

• Likewise, the reliability of the reaction time test was lower than that of other tests. It can be said that in environments where the player does not have support material, this situation increases the athlete's movement anxiety and reduces the reliability of the test

Figures/tables:

An additional figure showing the ‘shoulder joint proprioception sense’ test would be welcomed.

Table 4s legend says “p*<0.04. Should this be 0.05?

*Visual about joint position sense was added to the article. Images of other measurements can be added if desired.

• Error corrected, correct p*<0.05,

Table 1 mentions that the ‘*’ is for the ‘chi square test’. But this test is not mentioned in the methods. What is it adding to the Kruskal Wallis test?

Chi-square tests were performed to test the differences between shooting technique and the Cause of disability according to visual acuity level. Since shooting technique and Cause of disability are categorical variables, chi-square independence test is used.

Chi square test was used to compare categorical data with each other.

The Kruskal-Wallis test used to evaluate the change in numerical data such as joint position sense and reaction time according to visual acuity.

Line 228-230

6. PLOS authors have the option to publish the peer review history of their article (what does this mean?). If published, this will include your full peer review and any attached files.

yes

Do you want your identity to be public for this peer review? For information about this choice, including consent withdrawal, please see our Privacy Policy.

yes

---

## [Decision Letter · Decision Letter 1]

21 Dec 2023

Sensory function and somatosensorial system changes according to visual acuity and throwing techniques in goalball players: A cross-sectional study.

PONE-D-23-29101R1

Dear Dr. Gökşen,

We’re pleased to inform you that your manuscript has been judged scientifically suitable for publication and will be formally accepted for publication once it meets all outstanding technical requirements.

Kind regards,

Monika Błaszczyszyn

Academic Editor

PLOS ONE

Additional Editor Comments (optional):

Please consider the reviewer's minor suggestions 

Reviewers' comments:

Reviewer's Responses to Questions

**Comments to the Author**

1. If the authors have adequately addressed your comments raised in a previous round of review and you feel that this manuscript is now acceptable for publication, you may indicate that here to bypass the “Comments to the Author” section, enter your conflict of interest statement in the “Confidential to Editor” section, and submit your "Accept" recommendation.

Reviewer #1: All comments have been addressed

Reviewer #2: All comments have been addressed

2. Is the manuscript technically sound, and do the data support the conclusions?

Reviewer #1: Yes

Reviewer #2: Yes

3. Has the statistical analysis been performed appropriately and rigorously? 

Reviewer #1: No

Reviewer #2: Yes

4. Have the authors made all data underlying the findings in their manuscript fully available?

Reviewer #1: Yes

Reviewer #2: Yes

5. Is the manuscript presented in an intelligible fashion and written in standard English?

Reviewer #1: Yes

Reviewer #2: Yes

6. Review Comments to the Author

Reviewer #1: Dear Authors,

Thank you for the corrected version. Mostly my comments were taken into account. Please, see in the attachment (yellow notes) my suggestions.

Reviewer #2: Great job in addressing my comments. I have no further edits or suggestions. The paper can be accepted, unless the other reviewer has additional concerns.

7. PLOS authors have the option to publish the peer review history of their article (what does this mean?). If published, this will include your full peer review and any attached files.

Reviewer #1: No

Reviewer #2: **Yes: **Dustin J Oranchuk

---

## [Editor Report · Acceptance letter]

14 Jan 2024

PONE-D-23-29101R1 

PLOS ONE

Dear Dr. Gökşen, 

I'm pleased to inform you that your manuscript has been deemed suitable for publication in PLOS ONE. Congratulations! Your manuscript is now being handed over to our production team.

Kind regards, 

on behalf of

Dr. Monika Błaszczyszyn 

Academic Editor

PLOS ONE